# Rifabutin-Loaded Nanostructured Lipid Carriers as a Tool in Oral Anti-Mycobacterial Treatment of Crohn’s Disease

**DOI:** 10.3390/nano10112138

**Published:** 2020-10-27

**Authors:** Helena Rouco, Patricia Diaz-Rodriguez, Diana P. Gaspar, Lídia M. D. Gonçalves, Miguel Cuerva, Carmen Remuñán-López, António J. Almeida, Mariana Landin

**Affiliations:** 1R+D Pharma Group (GI-1645), Strategic Grouping in Materials (AEMAT), Department of Pharmacology, Pharmacy and Pharmaceutical Technology, Faculty of Pharmacy, Universidade de Santiago de Compostela-Campus Vida, 15782 Santiago de Compostela, Spain; helena.rouco@rai.usc.es; 2Drug Delivery Systems Group, Department of Chemical Engineering and Pharmaceutical Technology, School of Sciences, Universidad de La Laguna (ULL), Campus de Anchieta, 38200 La Laguna (Tenerife), Spain; pdiarodr@ull.edu.es; 3Research Institute for Medicines (iMed.ULisboa), Faculty of Pharmacy, Universidade de Lisboa, 1649-003 Lisbon, Portugal; diana.gaspar89@gmail.com (D.P.G.); lgoncalves@ff.ulisboa.pt (L.M.D.G.); aalmeida@ff.ulisboa.pt (A.J.A.); 4Department of Physical Chemistry, Nanomag laboratory, Universidade de Santiago de Compostela-Campus Vida, 15782 Santiago de Compostela, Spain; miguelcvidales@gmail.com; 5Nanobiofar Group (GI-1643), Department of Pharmacology, Pharmacy and Pharmaceutical Technology, Faculty of Pharmacy, Universidade de Santiago de Compostela-Campus Vida, 15782 Santiago de Compostela, Spain; mdelcarmen.remunan@usc.es

**Keywords:** rifabutin, nanostructured lipid carriers, cell uptake, Caco-2 cells, oral administration, Crohn’s disease

## Abstract

Oral anti-mycobacterial treatment of Crohn’s disease (CD) is limited by the low aqueous solubility of drugs, along with the altered gut conditions of patients, making uncommon their clinical use. Hence, the aim of the present work is focused on the in vitro evaluation of rifabutin (RFB)-loaded Nanostructured lipid carriers (NLC), in order to solve limitations associated to this therapeutic approach. RFB-loaded NLC were prepared by hot homogenization and characterized in terms of size, polydispersity, surface charge, morphology, thermal stability, and drug payload and release. Permeability across Caco-2 cell monolayers and cytotoxicity and uptake in human macrophages was also determined. NLC obtained were nano-sized, monodisperse, negatively charged, and spheroidal-shaped, showing a suitable drug payload and thermal stability. Furthermore, the permeability profile, macrophage uptake and selective intracellular release of RFB-loaded NLC, guarantee an effective drug dose administration to cells. Outcomes suggest that rifabutin-loaded NLC constitute a promising strategy to improve oral anti-mycobacterial therapy in Crohn’s disease.

## 1. Introduction

Crohn’s disease (CD) is a chronic inflammatory bowel condition with a higher predominance in industrialized countries, principally in Western Europe and North America [1]. The disease is characterized by the presence of outbreaks followed by remission periods [1,2], and although symptomatology is variable, diarrhea, abdominal pain, nausea, vomiting, and weight loss are usually involved [1]. The inflammatory process is usually transmural, involving any region of the digestive tract, affecting distal ileum and colon mainly [1,2].

CD aetiology has been a controversial topic recently [3]. Disease development is currently associated with genetic susceptibility and environmental factors, such as alterations in gut microbiome and treatment with antibiotics or non-steroidal anti-inflammatories [1,2]. Nevertheless, it is necessary to highlight the recent increment in scientific literature showing the contribution of the mycobacterial pathogen *Mycobacterium avium paratuberculosis* (MAP) in CD instauration [3,4]. Moreover, inflamed mucosal and submucosal layers in CD are infiltrated by immune cells such as macrophages [5]. These cells constitute an interesting target for anti-mycobacterial therapy, since MAP is a facultative intracellular organism that resides in host macrophages, establishing a persistent infection [6].

Despite this information, CD’s current treatment is still focused on the pharmacological control of the inflammatory process (using immunosuppressants, corticosteroids, anti-TNF or anti-interleukin drugs and adhesion molecule inhibitors) with the main objective of maintaining the disease remission stage without the need for surgery [1]. However, although these treatments improve patients’ quality of life, their ability to modify the long-term evolution of the disease has not been demonstrated yet [2].

Regarding the antibiotic use in CD, they are nowadays relegated to the treatment of perianal fistulas or disease suppurative complications [1]. Still, some case reports describe long-term CD remissions after antibiotic therapy [4]. Moreover, an open label extension phase III study sponsored by RedHill Biopharma is currently actively testing orally administered capsules containing a combination of rifabutin, clofazimine and clarithromycin at fixed doses in CD patients [7]. This study includes the introduction of a MAP PCR test at the baseline and the evaluation of changes of this blood status during the study [7], which would give insight into in vivo effectivity of this antibiotic combination [8] and into the clinical benefit derived from MAP eradication [9].

Although orally administered antimycobacterial drugs constitute a promising strategy in CD treatment, two aspects limit this approach. First, gut physiological parameters are altered in CD patients, which can reduce the possibilities to exploit pH, transit time or microbiome as targeting strategies for drug delivery [5]. On the other hand, antimycobacterial drugs show high lipophilicity and low oral bioavailability [10,11,12].

In this context, particulate systems constitute an interesting approach, as they can accumulate in inflamed bowel sites [5]. Additionally, nanoparticulated systems can be designed to load lipophilic drugs, improving their oral bioavailability [13,14]. Moreover, the drug particle reduction to nano size can lead to an enhanced water solubility and dissolution rate [15].

Among nanoparticulate systems, Nanostructured Lipid Carriers (NLC), the second generation of lipid nanoparticles (LN) [16], can be good candidates to formulate useful antimycobacterial systems. NLC are solid matrices at both room and body temperatures [17]. They are composed by a solid lipid and a liquid lipid [16] and present several advantages over the first generation of LN (known as Solid Lipid Nanoparticles or SLN), such as improved stability, higher suppleness in drug release modulation, and increased drug loading capacity [17]. NLC “in vitro” tolerability seems to be higher in comparison with other colloidal carriers, such as polymeric nanoparticles [18], making them an interesting option for oral drug administration.

Therefore, the aim of this work is to investigate the performance of rifabutin (RFB)-loaded NLC (whose formulation procedure and composition were previously optimized by Artificial Intelligence tools), to demonstrate their safety and suitability to achieve an appropriate intestinal permeability and an efficient macrophages uptake. Our goal is to improve the current Crohn’s disease treatments intended to eradicate MAP housed within intestinal macrophages, an area in which, to the best of our knowledge, nanotechnology has never been applied. In this way, an extensive characterization of the nanosystems in terms of particle size, polydispersity, surface charge, and drug payload, was performed. Thermal resistance, morphology, and drug release from NLC in different simulated media were also evaluated. Furthermore, an analysis of the in vitro performance of NLC in cell cultures including a permeability evaluation through Caco-2 monolayers, along with the assessment of cytotoxicity and uptake in human macrophages, was carried out in order to evaluate the targeting potential of the developed nanocarriers.

## 2. Materials and Methods

### 2.1. Materials

Rifabutin (RFB) (98% purity) was purchased from Acros Organics™ (Fair Lawn, NJ, USA). Polysorbate 80 (Tween^®^ 80), Coumarin 6, dialysis membrane (Spectrum™ Labs Spectra/Por, MWCO 3.5 KDa), and phorbol 12-myristate 13-acetate (PMA) were acquired from Sigma Aldrich (St Louis, MO, USA). Oleic acid was obtained from Merck (Darmstadt, Germany). Precirol^®^ ATO 5 (glyceryl distearate) and Epikuron^®^ 145 V (deoiled phosphatidyl choline-enriched lecithin) were kind gifts from Gattefossé (Saint-Priest, France) and Cargill (Wayzata, MN, USA) respectively. THP-1, Caco-2 human colon carcinoma and RAW 264.7 cell lines were obtained from ATCC (Manassas, VA, USA). Alexa Fluor™ 647 phalloidin, ProLong^®^ Gold Antifade reagent with DAPI, Gibco™ antibiotic-antimycotic (amphotericin B, penicillin, streptomycin), trypsin-EDTA, foetal bovine serum (FBS), Roswell Park Memorial Institute 1640 Medium (RPMI 1640), Minimum Essential Media (α-MEM), and phosphate buffered saline (PBS) were obtained from Thermo Fisher Scientific (Waltham, MA, USA). Dulbeco’s Modified Eagle Medium (DMEM) was purchased from Corning (Corning, NY, USA). Antibiotic solution (10.000 units/mL penicillin, 10.000 µg/mL streptomycin) was acquired from GE Healthcare Life Sciences (Chicago, IL, USA). Cell proliferation kit (WST-1) was purchased from Roche (Basel, Switzerland). Ultrapure water (MilliQ plus, Millipore Ibérica, Madrid, Spain) was used throughout and the remaining solvents and reagents were analytical or HPLC grade.

### 2.2. NLC Formulation

Several batches of NLC loaded with RFB were developed utilizing the composition and operating conditions of a previously optimized NLC system using Artificial Intelligence tools [19]. Briefly, the components of the formulation were Precirol^®^ ATO 5 and oleic acid as the lipid components (25:75 ratio), and Tween^®^ 80 and Epikuron^®^ 145 V as surfactants. The drug (15 mg) was dissolved in the molten lipid phase at 80 °C (300 mg). The aqueous phase (10 mL), a dispersion of Epikuron^®^ 145 V (0.5% *w*/*w* regarding the lipid phase weight) and Tween^®^ 80 (1.9% *w/v* regarding aqueous phase) in Milli-Q^®^ water, was heated at the same temperature, added to the lipid phase and hot shear homogenized (80 °C) using an Ultra-Turrax T25 (IKA Labortechnik, Staufen, Germany) at 14,800 rpm for 10 min, in a water bath. NLC dispersions were rapidly cooled by transferring them to an ice bath, with gentle stirring, for 2 min. Formulations were carried out in quintuplicate and subsequently dialyzed overnight (MWCO 3.5 KDa), in order to remove the non-incorporated components and obtain the purified NLC.

### 2.3. NLC Characterization

#### 2.3.1. Particle Size, Surface Charge and Physical Stability

Particle size, polydispersity index and surface charge of NLC were determined using a Zetasizer Nano ZS (Malvern Instruments, Malvern, UK). For size and polydispersity index determinations, samples were placed in polystyrene cuvettes and diluted with Milli-Q^®^ water (1:10). Surface charge was determined as zeta potential through particle mobility in an electric field. To carry out this determination, samples were also diluted with Milli-Q^®^ water (1:10) and placed in a specific cuvette where a potential of ±150 mV was established. All the measurements were performed at 25 ± 1 °C by quadruplicate.

#### 2.3.2. Transmission Electron Microscopy (TEM)

Transmission electron microscopy was employed to evaluate morphology of blank (control NLC without drug) and RFB-loaded NLC and to confirm particle sizes previously obtained by DLS. Thus, NLC suspensions were placed on formvar/carbon-coated grids (400 mesh) and stained with 2% (*w*/*v*) uranyl acetate. Finally, samples were analysed using a JEOL microscope (JEM 1010, Tokyo, Japan). Images were then obtained by using a CCD Orius-Digital Montage Plug-in camera (Gatan, Inc., Pleasanton, CA, USA) and analysed by means of a Gatan Digital Micrograph software (Gatan, Inc., USA). The number of particles considered for size determinations were 44 and 12 for blank and loaded NLC, respectively.

#### 2.3.3. Atomic Force Microscopy (AFM)

NLC morphology, particle size and size distribution were also analysed by atomic force microscopy (AFM). This technique is based on the electrostatic interaction between the sample and the AFM tip, which allows for the determination of a sample topography. Measurements were conducted under normal ambient conditions using an XE-100 instrument (Park Systems, Suwon, South Korea) in non-contact mode with the high-resonance non-contact AFM cantilever (ACTA probe, n = 330 kHz). For AFM imaging, 20 µL of the sample were dropped onto freshly exfoliated mica sheet (SPI Supplies, grade V-1 Muscovite) and after 5 min the mica was washed with Milli-Q water and dried under nitrogen flow. All experiments were performed at room temperature. XEI^®^ data processing tool (Park Systems, South Korea) were used for the analysis of the obtained data, which were adjusted to a gaussian distribution.

#### 2.3.4. Encapsulation Efficiency and Drug Loading

Encapsulation efficiency and drug loading determinations were performed as previously described [19]. Purified NLC and non-purified NLC (200 µL) were dissolved with acetonitrile (1.5 mL) and centrifuged at 16,099× *g* and 4 °C for 30 min. Centrifugation produces the precipitation of the lipid phase, while the drug remains in the supernatant. RFB quantification was performed by High Performance Liquid Chromatography (HPLC) as described in Section 2.4. The amount of drug quantified in the supernatant of non-purified nanoparticles was used as total drug content.

Encapsulation efficiency (EE) and drug loading (DL) of NLC were calculated using the following equations:EE (%) = [(W_loaded drug_)/W_total drug_] × 100,(1)
DL (%) = [(W_loaded drug_)/W_lipid_] × 100,(2)
where W_loaded drug_ is the amount of drug successfully incorporated in the formulation (remaining in the supernatant following acetonitrile addition), W_total drug_ is the total amount of drug, and W_lipid_ is the weight of the lipid vehicle.

#### 2.3.5. Thermal Analysis Using Dynamic Light Scattering (DLS)

The influence of temperature on both blank and RFB-loaded NLC suspensions stability was analysed by DLS in a Zetasizer Nano ZS. Three batches of each type of NLC (blank and loaded with RFB) were diluted as described above, and particle size measurements were made during heating and cooling cycles (25 °C-90 °C-25 °C) at 0.5 °C/min in quartz cells. Particle size determinations were recorded every 0.5 °C. Each batch was analysed in duplicate.

#### 2.3.6. In Vitro Release Studies

RFB release from loaded NLC was investigated in simulated intestinal fluid (SIF) and macrophage’s lysate, in order to compare NLC behaviour in different environments, the intestinal tract and inside macrophages. SIF with pancreatin was prepared according to United States Pharmacopeia (USP).

In order to obtain macrophages cell lysate, Raw 264.7 cells (a murine macrophage cell line) were cultured in DMEM supplemented with 10% foetal bovine serum (FBS) and 1% penicillin/streptomycin and incubated at 37 °C and 5% CO_2_. Cells were split when reaching 80% confluence by trypsinization and expanded until achieving enough number of cells. Cells were then trypsinized using standard conditions, washed with PBS, centrifuged, and resuspended in Milli-Q^®^ water in order to achieve a concentration of 3.125 million cells/mL. Cell lysis was performed by subjecting the cell suspension to three freeze-thaw cycles.

Drug release studies were performed by quadruplicate at 37 °C in horizontal Franz diffusion cells, where a 1:3 dilution of the nanoparticle suspension in release medium was put in the donor chamber. A dialysis membrane (MWCO 3.5 KDa) was placed between the two chambers in order to avoid the presence of NLC in the receptor chamber. At pre-set times, samples were taken from the receptor chamber and replaced with fresh medium. Drug quantification was performed by HPLC.

### 2.4. High Performance Liquid Chromatography Method

RFB was quantified following a validated method previously described [20], using an Agilent 1100 HPLC system (Agilent Technologies, Santa Clara, CA, USA) equipped with a C18 column (Waters symmetry 5 µm, 3.9 × 150 mm). Throughout HPLC analysis, 20 µL of each sample were injected and eluted with a mobile phase composed by a mixture of sodium acetate 0.05 M/potassium dihydrogen phosphate 0.05 M (pH adjusted to 4.0 with acetic acid) and acetonitrile (Scharlau, Barcelona, Spain) in a 53:47 (*v*/*v*) proportion. Drug quantification was performed at 275 nm, with a 1 mL/min flow rate in an isocratic mode.

### 2.5. In Vitro Cell Studies

#### 2.5.1. Cell Viability Studies

Cytotoxicity of NLC formulations was analysed using WST-1 (2-(4-iodophenyl)-3-(4-nitrophenyl)-5-(2,4-disulfophenyl)-2*H* tetrazolium, monosodium salt; Roche, Indianapolis, IN, USA), which produces a water-soluble formazan dye upon cellular reduction by the mitochondrial succinate-tetrazolium reductase [21,22]. Human monocytes (THP-1) were cultured in RPMI 1640 supplemented with 10% heat-inactivated foetal bovine serum (FBS), 1% penicillin/streptomycin and 2-mercaptoethanol 0.05 mM at 37 °C and 5% CO_2_. Five days before the experiment, cells were differentiated to macrophages by stimulation with 200 nM of PMA (Phorbol 12-myristate 13-acetate) for 3 days at a cell density of 2 × 10^5^ cells/mL. Then, PMA-containing medium was replaced by fresh medium and cells were incubated for another 2 days with normal media. The day before the experiment, cells were seeded at a density of 2.5 × 10^4^ cells/well in 96-well plates. Purified NLC samples were diluted to achieve a final concentration of 0.3, 0.12, 0.06, and 0.03 mg/mL of nanoparticles solid mass per volume. To evaluate cell viability, macrophages were incubated with blank and RFB-loaded NLC formulations, as well as with RFB solutions (concentration equivalent to those present in the previous NLC dilutions), for 24 h (37 °C, 5% CO_2_). After the incubation period, samples were removed and 10 µL of WST-1 reagent along with 100 µL of culture medium were added to each well. After 2 h of incubation with WST-1 reagent, absorbance was read at 450 nm in a microplate reader (Model 680, Bio-Rad, Hercules, CA, USA). Cell viability relative to negative control (Milli-Q^®^ water or DMSO, as appropriate) was calculated according to the following equation:Cell viability (%) = (Sample Absorbance/Control Absorbance) × 100,(3)

#### 2.5.2. Confocal Microscopy

Qualitative analysis of NLC internalization by THP-1 derived macrophages was performed by confocal microscopy. For this purpose, nanoparticles were fluorescently labelled with coumarin 6 by incorporating the fluorophore into the oil phase during the formulation process. Cells were seeded at a concentration of 5.3 × 10^4^ cells/cm^2^ in chambered cell culture slides (Nunc™ Lab-Tek II Chamber Slide™, Thermo Fisher Scientific, Waltham, MA, USA) the day before the experiment. Then, cells were incubated at 37 °C and 5% CO_2_ for 5 h with the samples (blank and RFB-loaded NLC), which were added in a final concentration of 0.12 mg/mL. After this incubation period, culture medium was removed, and cells were washed twice with pre-warmed phosphate buffered saline (PBS). Cell fixation was performed using a 3.7% formaldehyde solution in PBS for 10 min at room temperature, followed by two washing steps with PBS. Then, a 0.1% Triton X-100 solution was added to permeabilize the cell membrane. Finally, cells were incubated with a 1:40 dilution of Alexa Fluor™ 647 phalloidin in PBS for 20 min in order to label the macrophages cytoskeleton, and after two extra washing steps, macrophages nucleus were stained with ProLong^®^ Gold Antifade reagent with DAPI. Images were obtained using a confocal laser microscopy Leica SP5 (Leica Microsystems, Wetzlar, Germany).

#### 2.5.3. Macrophage Uptake Quantification

To quantify NLC uptake by THP-1 derived macrophages, NLC were fluorescently labelled with coumarin 6 as previously described. Dialysis of the samples was also accomplished prior to performing the experiment. Macrophage uptake quantification was carried out according to a method previously described [23] with slight modifications. First, macrophages were seeded in 96-well plates at a cell density of 2.5 × 10^5^ cells/mL and 100 µL per well; nanoparticle suspensions were added to them at a final concentration of 0.12 mg/mL, and fluorescence was determined in a microplate reader (Fluostar Optima, BMG Labtech, Offenburg, Germany) at an excitation and emission wavelength of 485 and 520 nm, respectively (Initial fluorescence). Cells were then incubated during 2 h at 37 °C and 5% CO_2_. Samples were removed, and cells were subjected to three washing steps with 250 µL of a 20 mM glycine solution in PBS pH 7.4, in order to remove non-internalized nanoparticles and to quench their fluorescent signals. Finally, 100 µL of Triton X-100 1% were added to disrupt cellular membrane, and fluorescence was again measured (Fluorescence post-lysis). Macrophage uptake was calculated according to the following equation:Macrophage uptake (%) = (Fluorescence post-lysis/Initial fluorescence) × 100,(4)

#### 2.5.4. Nanoparticle Permeation across Caco-2 Cells Monolayers

Permeation studies were performed in human colon carcinoma Caco-2 cell line according to a previously described protocol [24], with modifications. Cells were seeded at a concentration of 6.25 × 10^3^ cells/cm^2^ in Corning^®^ Transwell^®^ polycarbonate membrane cell culture inserts (Corning, Corning, NY, USA) and cultured in α-MEM supplemented with 20% FBS, 1% penicillin/streptomycin and 1% antibiotic/antimycotic. Culture medium was replaced every 3–4 days and cells were incubated at 37 °C and 5% CO_2_ for 28 days, approximately, until the monolayer reached a suitable transepithelial electrical resistance (TEER). At the beginning of the experiment, TEER was higher than 400 Ω cm^2^, which indicates the formation of an intact monolayer [24].

RFB-loaded NLC fluorescently labelled with coumarin 6 at a final concentration of 0.12 mg/mL or pure Milli-Q^®^ water (control), were added in the donor compartment. After 2, 4, 6, 24, and 48 h, samples were taken from the receptor compartment and replaced by fresh medium. Fluorescence was measured in a microplate reader (Fluostar Optima, BMG Labtech, Germany), as previously described, in order to evaluate NLC passage across the cell monolayer. Moreover, in order to correlate the amount of NLC present in the receptor compartment with the fluorescent signal obtained, a calibration curve was prepared in triplicate by measuring the fluorescence of known amounts of coumarin 6-labelled RFB-loaded NLC. Finally, permeability of NLC across Caco-2 cells was expressed either as the concentration of permeated RFB (µg/mL) regarding time elapsed or as a function of the apparent permeability coefficient (P_app_), which is employed to investigate the transport rate. P_app_ was determined according to the following equation:Papp (cm/s) = dQ/dt × 1/(A ×C_0_),(5)
where C_0_ is the initial RFB concentration in the upper compartment (6 µg/mL), A is the growth area (0.33 cm^2^) and dQ/dt is the appearance rate of the particles on the lower chamber based on its cumulative transport for 48 h. This linear appearance rate was calculated as the slope resulting from the representation of the RFB amount present in the receptor compartment versus time.

### 2.6. Statistical Analysis

All experiments were performed at least in triplicate. The data were expressed by mean ± SD and treated with IBM SPSS 24 software. The confidence interval was 95% (*p* ≤ 0.05). The groups were compared by performing one-way or two-way analysis of variance (ANOVA), as appropriate, followed by post hoc Tukey’s Multiple Comparison Test, and the significant differences between groups were determined.

## 3. Results and Discussion

### 3.1. NLC Characterization

NLC formulation procedure and composition were beforehand optimized by Artificial Intelligence (AI) tools in order to achieve optimal physicochemical properties along with a suitable drug payload [19]. Stability of the developed nanocarriers has proven to be adequate after 1 month of storage at 5 ± 1 °C, in terms of particle size, polydispersity index and drug payload. Minor changes without impact over colloidal stability were found for zeta potential [19]. Besides, an estimation of the characteristics of RFB-loaded NLC, prepared with these optimized parameters, was also provided [19]. In this way, to verify the robustness of this optimizations process, RFB-loaded NLC were prepared, and particle size, PDI, ZP, and drug payload were again determined. Furthermore, this work includes further characterization of these nanocarriers in terms of morphology, thermal behaviour, release profile, and in vitro performance in cell cultures.

#### 3.1.1. Particle Size, Surface Charge, Physical Stability, and Drug Payload

Blank and RFB-loaded NLC were prepared using hot high shear homogenization. Formulations were carried out in quintuplicate, dialyzed overnight and fully characterized in terms of particle size, size distribution, surface charge, and drug load (Table 1).

Particle size and size distribution are known to affect NLC characteristics such as stability, release rate and biologic performance [17], and because of that, they should be carefully characterized. NLC formulations showed particle sizes within the nano range, with values of 111 ± 3 nm and 151 ± 34 nm, for blank and RFB-loaded nanocarriers, respectively. Differences in size observed between blank and loaded formulations could be associated with the required accommodation space for the drug [25]. Regarding particle size distribution, blank NLC showed a polydispersity index (PDI) value of 0.23 ± 0.00, whereas the loaded ones displayed an almost identical value of 0.22 ± 0.02. Therefore, the obtained PDI values were below 0.3 in both cases, which is an acceptable value for lipid nanocarriers and indicative of homogeneous particle size distribution [26]. Remarkably, both size and PDI values obtained for RFB-loaded NLC are in close agreement with those previously predicted by Artificial Intelligence tools, which have been reported to be 152 nm and 0.23, for size and PDI, respectively [19].

Moreover, both blank and loaded nanocarriers showed zeta potential values close to −25 mV (−26 ± 2 and −24 ± 2 mV, for blank and RFB-loaded NLC, respectively), which guarantees a good colloidal stability if emulsifiers are included among formulation components [17,27]. These results differ slightly from those predicted by Artificial Intelligence tools, which showed slightly less negative values (−19 mV) [19]. However, these differences in ZP could be easily associated with the dialysis step performed in this work after NLC formulation, which can favour the removal of NLC superficial components, such as Tween^®^ 80, a non-ionic emulsifier. Since this type of emulsifier has been reported to localize close to the nanoparticle interface, counteracting the negative charge of the lipid matrix [28], its partial removal is expected to lead to a more negative zeta potential. Furthermore, these small differences could only have a slight impact on colloidal stability and are not likely to influence the in vivo fate of the nano-formulations.

Concerning drug payload, RFB was incorporated to NLC at 5% (*w*/*w*) regarding lipid matrix weight showing a suitable drug payload, with an encapsulation efficiency (EE) of 92.83 ± 3.75% and a drug loading (DL) of 4.62 ± 0.33%. These values suggest that almost all the added drug was successfully incorporated into the nanoparticle matrix. In the same way as in the case of particle size and PDI, EE and DL values obtained are almost identical to those predicted by Artificial intelligence tools, which have been reported to exhibit values of 100% and 5%, for EE and DL, respectively [19].

Hence, the NLC physicochemical characterization data show that they have a particle size within the nano-range, a monodisperse particle size distribution and a suitable drug payload. Besides, the highly negative zeta potential exhibited by the formulations is expected to promote a good colloidal stability. Finally, the results of RFB-loaded NLC characterization closely agree with those predicted by Artificial Intelligence, demonstrating the suitability of these tools to successfully optimize the design of nanoparticle-based drug delivery systems and develop robust and reproducible protocols of NLC preparation.

#### 3.1.2. Thermal Analysis Using Dynamic Light Scattering (DLS)

To assess NLC thermal stability, blank and RFB-loaded formulations were subjected to a heating stage from 25 °C to 90 °C, followed by a cooling step to the initial temperature. This approach was previously described to investigate the ability of lipid nanoparticle formulations to maintain their initial properties during high temperature-related procedures [23,29].

In the case of blank NLC (Figure 1A), particle size remains almost unchanged during the whole thermal analysis. A similar behavior was observed for RFB-loaded NLC (Figure 1B) but showing a slight reduction in nanoparticle size. Particle size maintenance along with the negligible size variations exhibited by both formulations throughout the assay indicate a good thermal stability. Therefore, results obtained suggest the developed NLC formulations are suitable for further temperature-requiring processes, as is the case of spray-drying [23], that could simplify the oral administration of NLC obtaining dried powders, which can be easily administered in capsules or tablets [30].

#### 3.1.3. Transmission Electron Microscopy (TEM)

TEM technique was employed to evaluate both blank and RFB-loaded NLC morphology as well as to verify nanocarriers size, as recommended elsewhere [27]. As shown in Figure 2, NLC exhibit a spheroidal morphology. Furthermore, in some images (such as Figure 2B), a structure with concentric layers could be noticed, which is also disturbed towards the center of the nanoparticle, exhibiting a high electron density. This lipid nanoparticle structure has been previously described and is associated with the polymorphic α-form of lipids [31]. Moreover, a size of 119 ± 41 nm in the case of blank NLC and slightly higher (173 ± 85 nm) in the case of RFB-loaded ones was observed, confirming the results obtained by DLS.

#### 3.1.4. Atomic Force Microscopy (AFM)

Blank and RFB-loaded NLC morphology, particle size and distribution were also assessed by Atomic force microscopy, a technique which gives insight into the sample z-dimension from the deflection of a fine leaf spring (known as the AFM cantilever) [32]. Therefore, AFM is a useful tool to complete the information obtained by DLS and by the two-dimensional images provided by TEM.

In this way, the AFM images of blank and RFB-loaded nanoparticles depicted in Figure 3 confirm the spheroidal shape previously shown by TEM. Moreover, results derived from AFM analysis were expressed as the frequency (%) of nanoparticles exhibiting a specific height. Thus, blank and RFB-loaded nanoparticles exhibited a similar size, as values of 43 ± 3 nm and 33 ± 1 nm, respectively, were obtained (Figure 4). The smaller nanoparticle height reported by AFM in comparison with the diameters obtained by DLS and TEM corroborates the existence of a spheroidal structure, closer to a disk than to a sphere. This structure further confirms the prevalence of the polymorphic α-form of lipids [27,31], as previously mentioned, which has been associated with a high loading capacity and a low tendency to expulse the encapsulated drug from the lipid matrix [27].

#### 3.1.5. In Vitro Release Studies

Release studies were performed to investigate the ability of NLC to act as RFB reservoirs. This compound is an antimicrobial agent active against widely known *Mycobacterium* species, including *M. leprae*, *M. tuberculosis*, or MAP [8], an infectious agent increasingly related with CD development [3]. Furthermore, RFB has been included in the triple oral anti-mycobacterial regimen currently under evaluation intended to eradicate MAP infection from CD patients [8]. In this way, release from nanoparticles was performed in both simulated intestinal fluid (SIF) and macrophage’s lysate, with the aim of analyzing the expected drug release profile in the gut and intracellular environment, respectively. In the drug release study in SIF, the experimental results indicated that the RFB release from NLC in intestinal environment is negligible, since the amount of drug released during the whole assay was not detected by HPLC. Regarding RFB release assay in murine macrophage’s lysate, the released drug started to be detectable at 1 h, and increased progressively until 16 h, when the amount of RFB in release medium was found to be quantifiable by HPLC. After 16 h, 1.46 ± 0.47 µg of drug were detected in the receptor chamber of Franz diffusion cells, resulting in a release percentage of 0.1 ± 0.03% at the end of the experiment.

In physiological conditions, drug release from lipid nanocarriers has been reported to occur, simultaneously, through erosion and diffusion mechanisms [33]. Poor RFB release from NLC obtained in the assays is likely to be related with the high lipophilicity of the drug (log P = 4.218) [34], as well as with the favorable conditions within the nanoparticle due to the high RFB solubility in the lipid matrix, which significantly reduces drug diffusion towards the aqueous release medium [35]. Several authors employed different strategies to overcome these issues associated with the poor in vitro release profile of lipophilic drugs, such as the addition of ethanol [36] or surfactants [37] in the release medium. However, in most cases, these approaches are not representative of the in vivo environment.

Regarding drug release through lipid matrix degradation, it is necessary to consider that it occurs primarily by enzymes, and also through hydrolytic processes, although to a lesser extent [33]. Because of that, analysis of NLC matrix degradation by enzymes present in both intestine and macrophage intracellular environment constitutes an interesting approach. According to the results obtained, NLC matrix can efficiently endure pancreatin activity of SIF, however, it is more affected by enzymes present in macrophage intracellular environment. Despite the higher effectivity of these macrophage enzymes, the amount of drug released from nanoparticles is still low. This slight drug release can be associated with the low enzymatic concentration in macrophage lysate achieved after dilution of cell suspension with Milli-Q^®^ water. Therefore, as this concentration would obviously be higher inside macrophages in physiological conditions, a greater drug release would also be expected in this case.

Hence, outcomes obtained constitute an interesting proof of concept of the controlled and selective drug release provided by NLC inside macrophages, where MAP has been reported to establish a persisting infection [6].

### 3.2. In Vitro Cellular Studies

#### 3.2.1. Cell Viability Studies

SLN and NLC are composed of biodegradable lipids with generally recognised as safe (GRAS) status [38]. However, despite these promising features, further studies are required to support their therapeutic use [18]. For this purpose, cell viability of THP-1 derived macrophages was analysed after NLC treatment.

Figure 5a shows cell viability after treatment with blank and RFB-loaded formulations at several concentrations (0.3, 0.12, 0.06 and 0.03 mg/mL). Besides, cells treated with equivalent RFB concentrations were used as control (Figure 5b). In general, formulations exhibited a good biocompatibility, leading to cell viabilities ≥70% for concentrations lower than 0.3 mg/mL. Two-way ANOVA (*p* ˂ 0.05) points out a statistically significant effect of treatment (blank NLC or RFB-loaded NLC), NLC concentration, and their interaction on cell viability.

Post hoc Tukey’s Multiple Comparison Test (*p* ˂ 0.05) points out a NLC concentration-dependent cytotoxic effect. However, it shows no statistical differences in cell viability for the experiments carried out with RFB-loaded NLC at 0.12 mg/mL and 0.06 mg/mL, therefore, the concentration 0.12 mg/mL was selected for further assays.

On the other hand, no significant modifications in cell viability were observed for RFB solutions (Figure 5b) containing equivalent amounts of drug, which implies that the reduction in viability must be mainly attributed to the toxicity of NLC components (oleic acid or emulsifiers), as suggested by several authors [39,40], and not to cytotoxic effects derived from RFB inclusion in the nanocarriers.

To allow the comparison of the cytotoxicity results obtained for RFB-loaded NLC with available data in literature, determination of IC_50_ was accomplished (Figure 6). The estimation of this value was performed from dose-response curves, where IC_50_ was defined as the concentration required to induce a 50% reduction in cell viability, resulting in a value of approximately 0.18 mg/mL. This finding is in line with previously published studies on lipid nanoparticle cytotoxicity, where IC_50_ was reported to be found mainly in the range of 0.1–1 mg/mL nanoparticles [18].

From the results obtained, NLC formulations present a suitable safety profile, similar to previously published works with lipid nanoparticles. Moreover, the highest NLC concentration of all the tested ones, showing an adequate biocompatibility in THP-1 derived macrophages, has been found to be 0.12 mg/mL. A cell viability ≥ 70% has been reported to be the threshold, according to ISO 10993-5 [41], below which a cytotoxic effect is considered to take place.

#### 3.2.2. Confocal Microscopy

Qualitative study of both blank and RFB-loaded NLC internalization by THP-1 derived macrophages was analysed employing a nanoparticle concentration of 0.12 mg/mL, in accordance with the results obtained in the cell viability experiment. Moreover, the selection of this value was based not only on the results obtained in the biocompatibility assay, but also on the drug concentration required to efficiently eradicate the mycobacterial infection.

Regarding this last point, a minimum inhibitory concentration (MIC) ranging from 0.5 to 4 µg/mL has been obtained in vitro for RFB in both human and animal isolated MAP strains. Interestingly, five of six MAP human isolates show a MIC of only 1 µg/mL for this drug [42]. Furthermore, RFB has been reported to slow the multiplication of three virulent strains of Mycobacterium avium complex [43], a group of which MAP is an important member [44], in a model of intracellular infection in human macrophages when a dose of 0.5 µg/mL is employed [43]. Considering that RFB payload in NLC has been proven to reach almost 5% of the solid content of the formulations, a dose of approximately 6 µg/mL of drug was administered to macrophages. This drug concentration is clearly superior to the value required for RFB MIC, guaranteeing the administration of an effective dose to cells.

As observed in Figure 7, both blank (Figure 7A) and RFB-loaded NLC (Figure 7B) have been efficiently taken up by macrophages, which constitutes a promising start point for the treatment of infections produced by intracellular pathogens such as MAP. Images of separate channels of blank and loaded formulations are shown in Appendix A, respectively. Furthermore, images of NLC uptake by a macrophages group can be found in Appendix A.

#### 3.2.3. Macrophage Uptake Quantification

Nanoparticle internalization by macrophages was also investigated using a quantitative approach and NLC formulations at 0.12 mg/mL. In this method, blank formulations showed an internalization percentage of 8.33 ± 1.15% after 2 h of exposure. Moreover, a higher internalization percentage, 13.39 ± 1.44%, was obtained for RFB-loaded NLC. Statistical analysis (one-way ANOVA, *p* ˂ 0.05) revealed significant differences between the uptake of blank and loaded formulations. This higher macrophages uptake reported for loaded formulations might be related to their size (151 ± 34 nm), larger than the blank ones (111 ± 3 nm), since the uptake of particulate systems by macrophages rises progressively with the particle size increase [45].

In addition, the uptake percentage obtained for RFB-loaded NLC reveals that 13% of the administered dose (6 µg/mL or 0.6 µg each well) was efficiently taken up by macrophages, which constitutes a total RFB amount of 0.078 µg per well. Since each well contains 25.000 cells and human macrophages have a cell volume of approximately 4990 µm^3^ [46], a total cell volume of 1.2475 × 10^−4^ mL is expected. Taking together the volumes of internalized drug and the total cell volume estimated, it is reasonable to think that an internalized concentration of 625 µg/mL could be achieved, which exceeds the range of the intracellular MIC previously reported. It is also important to note that colocalization phenomena amongst nanoparticles and mycobacteria have been suggested to occur by means of phagolysosomes fusion [47]. In this way, RFB would not be free in the cytoplasm and hence, it would probably not be the substrate of efflux pumps, which could modify the intracellular drug concentration indicated in this work.

#### 3.2.4. Nanoparticle Permeation across Caco-2 Cells Monolayers

Nanoparticle permeation across the intestinal barrier is required to reach intestinal macrophages. Despite some of them being able to extend dendrites into the intestinal lumen, the majority of the macrophages are located below the epithelial monolayer, in the lamina propria [48]. The analysis of nanoparticles permeability across human colon carcinoma cell monolayers (Caco-2) has been reported to establish a good correlation with human in vivo absorption and has been broadly employed to predict drug permeability [24]. The drug concentration in the receptor compartment was estimated from the NLC concentration in this compartment at different times. Results are shown in Figure 8. No NLC permeation across the Caco-2 monolayer occurred during the first 2 h. After, RFB concentration in the receptor compartment increased linearly over time, achieving RFB concentrations of 3.29 × 10^−4^ 0.06 ± 0.05, 0.43 ± 0.01 and 0.9 ± 0.1 µg/mL at 4, 6, 24, and 48 h, respectively. In addition, a Papp value of 2.02 × 10^−6^ cm/s was obtained for RFB-NLC formulations. A Papp value of 2 × 10^−6^ cm/s has been reported as the threshold to achieve complete drug absorption in humans [49], therefore, the obtained value for RFB-NLC suggests that they exhibit good permeability across Caco-2 monolayers.

Besides, the achieved permeability allowed to obtain a drug concentration virtually in range with the previously mentioned MIC reported for RFB just after 24 h of incubation with the nanocarriers, which is the accurate colonic transit time described for patients with CD [5]. Moreover, the in vivo permeability exhibited by NLC across the intestinal membrane of CD patients is expected to be even higher than reflected by this assay. The disruption of the epithelial barrier shown by inflammatory bowel disease patients increases gut permeability and favors nanoparticle passage [5]. In this way, nanoparticles are expected to accumulate preferentially in the intestinal inflamed sites, which are densely infiltrated with macrophages [5], ensuring the administration of an effective drug dose to the infected cells.

## 4. Conclusions

NLC showing a particle size within the nano-range (111 ± 3–151 ± 34 nm); a monodisperse size distribution (0.22 ± 0.02–0.23 ± 0), a negative zeta potential (−24 ± 2–−26 ± 2 mV), and a suitable rifabutin payload (4.62 ± 0.33%) were successfully prepared using a formulation process previously optimized by Artificial Intelligence tools. Formulations exhibited a spheroidal appearance, a good ability to withstand high temperature-related processes and a good safety profile in cellular studies. Moreover, the efficient macrophage uptake of the developed NLC has been demonstrated allowing the obtention of a therapeutic rifabutin concentration after only 2 h of incubation. This fact, along with the permeation exhibited by NLC across Caco-2 cell monolayers and their tendency to release the drug in the intracellular environment, guarantee the achievement of an effective rifabutin dose inside the phagocytic cells, where mycobacterium avium paratuberculosis is known to reside. Therefore, rifabutin-loaded NLC constitute a promising tool to improve anti-mycobacterial therapy in Crohn’s disease.

## Figures and Tables

**Figure 1 nanomaterials-10-02138-f001:**
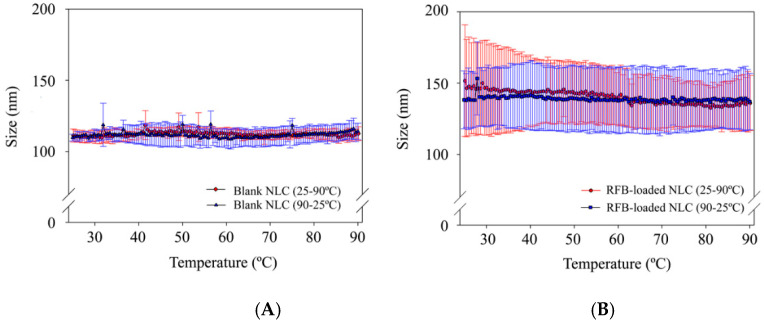
Dynamic light scattering thermograms of (**A**) Blank NLC formulations and (**B**) RFB-loaded NLC formulations.

**Figure 2 nanomaterials-10-02138-f002:**
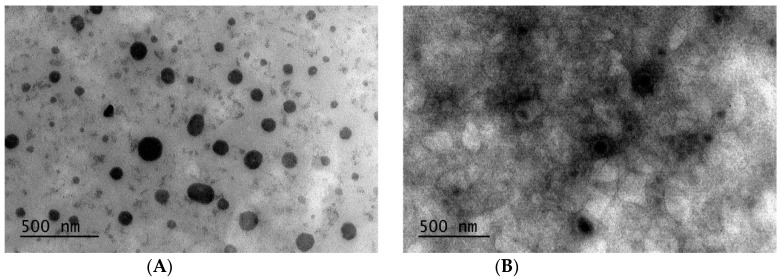
Transmission electron micrographs of (**A**) Blank and (**B**) RFB-loaded NLC.

**Figure 3 nanomaterials-10-02138-f003:**
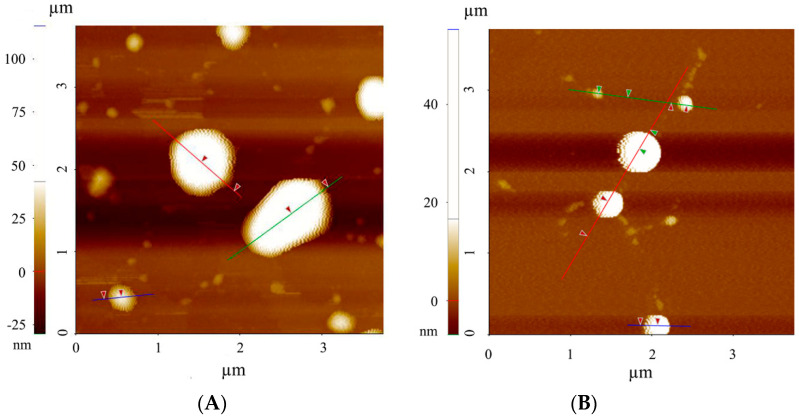
Atomic force microscopy images of (**A**) Blank and (**B**) RFB-loaded NLC.

**Figure 4 nanomaterials-10-02138-f004:**
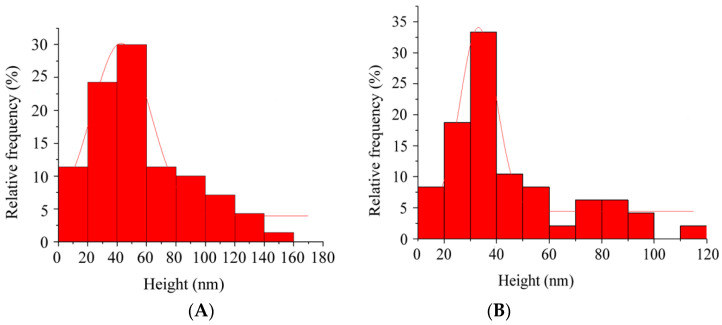
Particle size and size distribution according to AFM analysis of (**A**) Blank and (**B**) RFB-loaded NLC. Results were adjusted to a gaussian distribution.

**Figure 5 nanomaterials-10-02138-f005:**
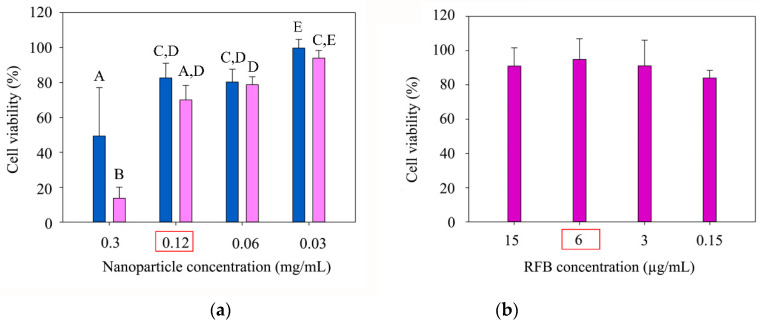
Cell viability (%) relative to control with Milli-Q^®^ water of (**a**) blank NLC formulations (dark blue colour) and RFB-loaded formulations (pink colour), using several nanoparticle concentrations (0.3, 0.12, 0.06 and 0.03 mg/mL). (**b**) Cell viability (%) relative to control with DMSO, of RFB solutions in DMSO prepared employing the same drug concentration as present in NLC formulations (purple colour). (A–E characters denote the homogeneous subsets pointed out by the post hoc Tukey’s Multiple Comparison Test (*p* ˂ 0.05).

**Figure 6 nanomaterials-10-02138-f006:**
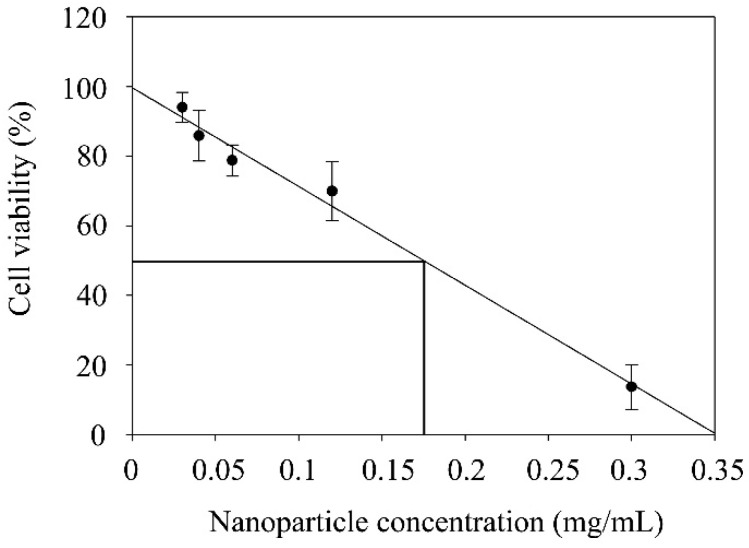
Dose response curve (R^2^ = 0.9881) obtained from cell viability data at several RFB-loaded NLC concentrations. IC_50_ was calculated as the concentration of nanoparticles producing a 50% reduction in cell viability.

**Figure 7 nanomaterials-10-02138-f007:**
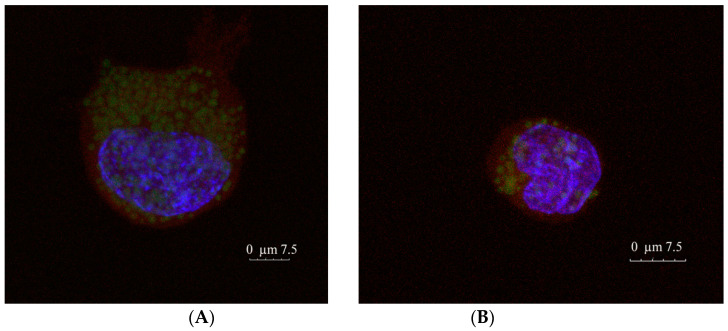
Confocal microscopy images of (**A**) Blank and (**B**) RFB-loaded NLC macrophages uptake. Red, blue, and green colours represent cell cytoplasm (Alexa Fluor™ 647 phalloidin), cell nuclei (DAPI) and NLC formulations (Coumarin 6), respectively.

**Figure 8 nanomaterials-10-02138-f008:**
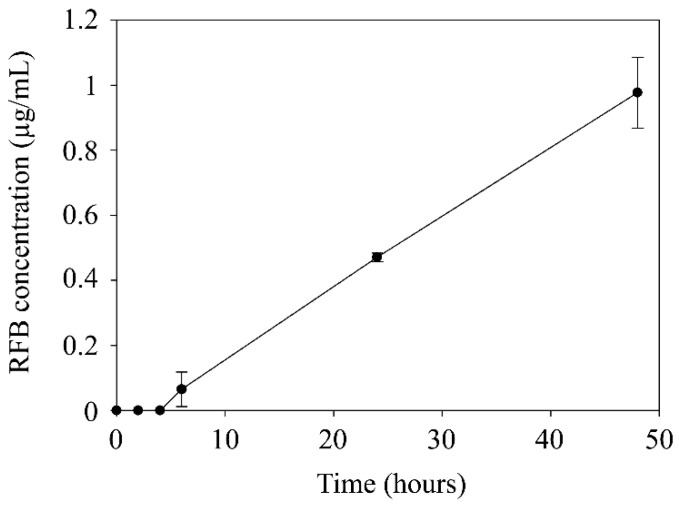
Permeation profile of rifabutin (RFB) loaded in NLC across Caco-2 cell monolayers expressed as a function of drug concentration in the lower compartment in a 48-h time interval.

**Table 1 nanomaterials-10-02138-t001:** Blank and RFB-loaded NLC characterization in terms of particle size, PDI, ZP, EE, and DL (n = 3 ± SD).

NLC	Size (nm)	PDI	ZP (mV)	EE (%)	DL (%)
Blank	111 ± 3	0.23 ± 0	−26 ± 2	-	-
RFB-loaded	151 ± 34	0.22 ± 0.02	−24 ± 2	92.83 ± 3.75	4.62 ± 0.33

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
