# Peer review of "Rifabutin-Loaded Nanostructured Lipid Carriers as a Tool in Oral Anti-Mycobacterial Treatment of Crohn’s Disease"

_nanomaterials, 2020, doi:10.3390/nano10112138_

Round 1
Reviewer 1 Report
General comments
The present manuscript reports on the production and characterisation of rifabutin-loaded nanostructured lipid carriers.
The paper is well organised and complete. The topic well fits the aim and scope of Nanomaterials and is interesting and worthy of investigations.
Some minor revisions have to be done, as reported below.
Abstract
The Abstract section has to be summarised in order to make it more immediate and incisive. Many details, particularly the numerical data, can be moved to the Conclusions section.
Keywords
The chosen keywords (i.e. Caco-2 cells; Cell uptake; Crohn´s disease; Nanostructured lipid carriers; Oral administration; Rifabutin.) have to be reported in a more logical order (i.e. material, processing, characterisations, properties, applications).
- Introduction
- The introduction is well organised, but, even if the aim is clear, the Authors should better evidence the originality of their work and the added value to the scientific knowledge about the considered topic.
- The following statement “Additionally, nanoparticulated systems can be designed to load
- 109 lipophilic drugs, improving its oral bioavailability” has to be supported with recent suitable references, including “Strategies To Improve Ellagic Acid Bioavailability: From Natural Or Semisynthetic Derivatives To Nanotechnological Approaches Based On Innovative Carriers, Nanotechnology 31[38] (2020): 382001” and “Controlled release of 18-β-glycyrrhetic acid by nanodelivery systems increases cytotoxicity on oral carcinoma cell line, Nanotechnology 29[28] (2018) 285101 (11pp).”.
- It is strongly suggested to add a brief list of the used characterisations at the end of the Introduction section.
- Results and Discussion
3.1.2. Thermal analysis using dynamic light scattering (DLS)
- The following consideration “Therefore, results obtained suggest the developed NLC formulations are suitable for further temperature-requiring processes, as is the case of spray-drying, that could simplify the oral administration of NLC obtaining dried powders, which can be easily administered in capsules or tablets.” needs suitable references.
3.1.3. Transmission electron microscopy (TEM)
- The average dimensions, as well as the related standard deviations, have to be added. Moreover, in the Experimental section, the Authors have to specify the used software for the size measurements, and the number of considered particles.
Author Response
We would like to thank the reviewers and editor for the careful reading of the manuscript as well as for the positive feedback given. We have tried to answer each concern (our responses in blue) and we think the manuscript has notably been improved.
Reviewer #1 comments:
Comment 1: The Abstract section has to be summarised in order to make it more immediate and incisive. Many details, particularly the numerical data, can be moved to the Conclusions section.
The abstract section has been modified as suggested by the reviewer. Please check the text highlighted in yellow. Moreover, numerical data were transferred to the conclusions section.
Comment 2: The chosen keywords (i.e. Caco-2 cells; Cell uptake; Crohn´s disease; Nanostructured lipid carriers; Oral administration; Rifabutin.) have to be reported in a more logical order (i.e. material, processing, characterisations, properties, applications).
Keywords order has been modified as follows: Rifabutin; Nanostructured lipid carriers; Cell uptake; Caco-2 cells; Oral administration; Crohn´s disease.
Comment 3: The introduction is well organised, but, even if the aim is clear, the Authors should better evidence the originality of their work and the added value to the scientific knowledge about the considered topic.
We agree. A new paragraph regarding this concern has been added. “Our goal is to improve the current Crohn´s disease treatments intended to eradicate MAP housed within intestinal macrophages, an area in which, to the best of our knowledge, nanotechnology has never been applied.”
Comment 4: The following statement “Additionally, nanoparticulated systems can be designed to load 109 lipophilic drugs, improving its oral bioavailability” has to be supported with recent suitable references, including “Strategies To Improve Ellagic Acid Bioavailability: From Natural Or Semisynthetic Derivatives To Nanotechnological Approaches Based On Innovative Carriers, Nanotechnology 31[38] (2020): 382001” and “Controlled release of 18-β-glycyrrhetic acid by nanodelivery systems increases cytotoxicity on oral carcinoma cell line, Nanotechnology 29[28] (2018) 285101 (11pp).”
We have found the suggested papers of great interest, and we have included them as references in the previous statement. Please check the highlighted text in the manuscript (introduction section).
Comment 5: It is strongly suggested to add a brief list of the used characterisations at the end of the Introduction section.
We have included a brief list of the characterization techniques employed in this work at the end of the introduction section. Please find the added text below:
“In this way, an extensive characterization of the nanosystems in terms of particle size, polydispersity, surface charge and drug payload, was performed. Thermal resistance, morphology, and drug release from NLC in different simulated media were also evaluated. Furthermore, an analysis of the in vitro performance of NLC in cell cultures including a permeability evaluation through Caco-2 monolayers, along with the assessment of cytotoxicity and uptake in human macrophages was carried out, in order to evaluate the targeting potential of the developed nanocarriers.”
Comment 6: The following consideration “Therefore, results obtained suggest the developed NLC formulations are suitable for further temperature-requiring processes, as is the case of spray-drying, that could simplify the oral administration of NLC obtaining dried powders, which can be easily administered in capsules or tablets.” needs suitable references.
According to the reviewer suggestion, the following references were added to the text: Gaspar, D.P.; Faria, V.; Goncalves, L.M.; Taboada, P.; Remunan-Lopez, C.; Almeida, A.J. Rifabutin-loaded solid lipid nanoparticles for inhaled antitubercular therapy: Physicochemical and in vitro studies. Int J Pharm 2016, 497, 199-209, Battaglia, L.; Gallarate, M. Lipid nanoparticles: state of the art, new preparation methods and challenges in drug delivery. Expert opinion on drug delivery 2012, 9, 497-508. Please check the highlighted text in the manuscript (section 3.1.2).
Comment 7: The average dimensions, as well as the related standard deviations, have to be added. Moreover, in the Experimental section, the Authors have to specify the used software for the size measurements, and the number of considered particles.
The software employed for the size measurements (Gatan Digital Micrograph software) and the number of considered particles were added in the experimental section. NLC average dimensions derived from TEM analysis, along with the standard deviations obtained have been included in the results and discussion section. Please, check the highlighted text in the manuscript (section 3.1.3).

Reviewer 2 Report
This manuscript reported the preparation of rifabutin-loaded nanostructured lipid carriers via hot homogenization. The resulting materials are expected to be used in the anti-mycobacterial treatment of Crohn’s disease by improving the drug solubility. It was found that those nanomaterials could offer excellent properties and unique advantages in potential biological applications. Although interesting, the results are not new and this study cannot really address the practical problems in this area. Therefore, I do not think this manuscript should be published by the journal of Nanomaterials. Some specific reasons are listed here as following:
- The major concern of this paper is lack of novelty. Indeed there is a number of articles reporting that Drugs-loaded NLC for the treatment of chronic inflammatory bowel condition during the past years. The authors should carefully clarify the unique contribution and significance of this work in the introduction part.
- This current version lacks key assessment in vivo including the assessment of drug in colon tissue and NLC therapy in vivo. Available data is also hard to prove NLC system can really improve the drug concentration and it can contribute to the therapy for Crohn`s.
- There is no data in the manuscript discussion the relationship between RFB-loaded drugs and the treatment of diseases. This article just indicates the macrophage uptake of the developed NLC. However,the data of figure 7 is not reasonable, which just show an individual cell,maybe a group of macrophages is more effective. Moreover, there is no a graph quantify the difference, so the result is not effective due to the not clear enough image.
- The explanation of the Anti-inflammatory mechanism is not very clear. This work is focusing on the improvement of the macrophage uptake of the drug, but the mechanism of macrophage is not enough. Also, the mechanism of RFB-loaded NLC is not explained on the cell level or the molecular level. A number of articles has reported that the mechanism of RFB and the molecular expression of macrophage during the anti-inflammatory process.
Author Response
We would like to thank the reviewers and editor for the careful reading of the manuscript as well as for the positive feedback given. We have tried to answer each concern (our responses in blue) and we think the manuscript has notably been improved.
Comment 1: The major concern of this paper is lack of novelty. Indeed there is a number of articles reporting that Drugs-loaded NLC for the treatment of chronic inflammatory bowel condition during the past years. The authors should carefully clarify the unique contribution and significance of this work in the introduction part.
We agree with the reviewer that nanoparticles have been extensively employed in the treatment of inflammatory bowel diseases (IBD). However, the novelty of this work does not rely neither on NLC as carriers of poorly soluble drugs, nor on the use of these nanoparticulated carriers in IBD. Instead, the originality and the added value of this manuscript relies of the use of NLC to improve the anti-mycobacterial therapy of Crohn´s disease. Promising results have been obtained in the treatment of Crohn's disease ulcers with triple antibiotic therapy (Rifabutin, clarithromycin, clofazimine) [8]. However, researchers have pointed out the problems associated with a low bioavailability of those antibiotics, particularly rifabutin and clofazimine, whose improvement should undoubtedly lead to greater success of this treatment. In this way, to the best of our knowledge, the use of nanoparticles to improve this therapeutic approach, has not been described so far. However, we agree with the reviewer that the introduction section does not reflect properly the novelty of this work, and therefore we have included the text below to clarify this point:
“Our goal is to improve the current Crohn´s disease treatments intended to eradicate MAP housed within intestinal macrophages, an area in which, to the best of our knowledge, nanotechnology has never been applied.”
Comment 2: This current version lacks key assessment in vivo including the assessment of drug in colon tissue and NLC therapy in vivo. Available data is also hard to prove NLC system can really improve the drug concentration and it can contribute to the therapy for Crohn`s.
Regarding drug assessment in colon tissue, indeed we consider this determination could be crucial in the evaluation of nanoparticles for intestinal epithelium local therapy, such as anti-inflammatory drugs. Nonetheless, the aim of this work is to assess nanoparticles permeability through the intestinal barrier, allowing them to be internalized by intestinal macrophages, the target of RFB-loaded NLC. In this way, permeation studies using Caco-2 monolayers, as the one performed in this work, have been widely employed to evaluate intestinal permeability. These studies present a strong correlation with in vivo absorption, providing, in this way, reliable information. On the other hand, concerning the general lack of in vivo studies in this manuscript, in the current initial stage of NLC formulations development, in vivo experimentation could be replaced by suitable in vitro studies, in line with the principle of the 3Rs (replacement, reduction, and refinement) that should guide animal experimentation. However, we agree with the reviewer that further steps would require in vivo assays.
Comment 3: There is no data in the manuscript discussion the relationship between RFB-loaded drugs and the treatment of diseases. This article just indicates the macrophage uptake of the developed NLC. However the data of figure 7 is not reasonable, which just show an individual cell maybe a group of macrophages is more effective. Moreover, there is not a graph quantify the difference, so the result is not effective due to the not clear enough image.
We agree with the reviewer with the need of discussing the relationship of RFB and the treatment of diseases. In this way, the following paragraph was added in the discussion (section 3.1.5.):
“Release studies were performed to investigate the ability of NLC to act as RFB reservoirs. This compound is an antimicrobial agent active against widely known Mycobacterium species, including M. leprae, M. tuberculosis, or MAP [8], an infectious agent increasingly related with CD development [3]. Furthermore, RFB has been included in the triple oral anti-mycobacterial regimen currently under evaluation intended to eradicate MAP infection from CD patients [8].”
In this work, confocal microscopy analysis was employed as a merely qualitative assay intended to graphically show NLC internalization. In this way, we agree with the reviewer with the impossibility of quantifying NLC internalization from the information gathered from this study. Moreover, we also think that, although a group of macrophages could offer a more significant information, this would still be limited. Because of that, instead of using confocal microscopy to quantify NLC internalization by macrophages, we propose to use of a complementary study, in which fluorescence is employed to quantify nanoparticle internalization. This assay constitutes a fast way of evaluating cell internalization, and also allowed us to use a high number of cells, increasing results reliability. Nevertheless, we have added a confocal microscopy image of a group of macrophages in the supplementary materials.
Comment 4: The explanation of the Anti-inflammatory mechanism is not very clear. This work is focusing on the improvement of the macrophage uptake of the drug, but the mechanism of macrophage is not enough. Also, the mechanism of RFB-loaded NLC is not explained on the cell level or the molecular level. A number of articles has reported that the mechanism of RFB and the molecular expression of macrophage during the anti-inflammatory process.
We agree with the reviewer with the need of performing further studies to determine the influence of RFB-loaded NLC on the immune activation of macrophages. However, considering RFB is an antimicrobial agent, and pathogenic mycobacteria subvert the innate immune response as a strategy to survive within phagocytic cells , it would be more adequate to assess the capacity of the developed nanocarriers to induce immunity in Mycobacterium-infected macrophages, as suggested in previous works (Upadhyay, T.K.; Fatima, N.; Sharma, A.; Sharma, D.; Sharma, R. Nano-Rifabutin entrapment within glucan microparticles enhances protection against intracellular Mycobacterium tuberculosis. Artificial Cells, Nanomedicine, and Biotechnology 2019, 47, 427-435). In this way, we could assess if this immune response would allow for the elimination of intracellular bacteria. However, this work is an initial step aimed to evaluate the potential of RFB-loaded NLC, and only healthy macrophages were employed.

Reviewer 3 Report
- The manuscript describes the preparation and characterisation of nanostructured lipid carriers (NLC) loaded with an antibiotic (rifabutin) in the treatment of Crohn's Disease (CD). There is evidence that Mycobacterium avium paratuberculosis (MAP) is an important aetiological agent in CD so treatment with rifabutin may improve pathology. Also - since MAP has been reported to evade the immune system by residing in phagosomes of intestinal macrophages, and nanoparticles are predominantly taken up by macrophages, nanoparticle formulated rifabutin may improve drug activity. Overall, the study has been well designed and performed. However, I still have some (largely technical) queries.
- Section 2.5.3 Macrophage uptake quantification. How/why does this method quantitatively measure "macrophage uptake"? Specifically, what do the authors mean by "uptake"? In normal usage in the field this would be expected to indicate internalisation, but from from the scant description of the assay in the methods section it seems that "cellular/macrophage association" is more likely to be what is being measured. Note that it was very difficult to understand exactly how the assay was performed based on the description in the manuscript. The authors also reference a previous paper from the group (ref 21) but the description in the referenced paper also lacked clarity/detail (for example - the sentence in ref 21 describing "initial amount/[initial fluorescence]" and an internalisation calculation is in garbled English and uninterpretable). Please include a more detailed description of the method and explain how if measures "uptake" rather than "cellular association".
- Section 3.1.5 (2.3.6). Experiments were performed to assess stability in the "intracellular environment" of the macrophage using freeze/thaw lysates of cultured macrophages. A whole cell lysate is not a good analogue for the environment that a nanoparticle taken up by a macrophage will encounter. The NLCs will almost certainly be taken up by phagocytosis and will therefore be located in phagolysosomes where the conditions are extreme compared to the general cytoplasmic environment (e.g. highly concentrated reactive oxygen/nitrogen intermediates, degradative enzymes, low pH). Is there any evidence that the RFB loaded into the NLC can survive these conditions in an active form?
- Section 3.2.2. Confocal microscopy was performed to examine the subcellular localisation of the RFB-loaded NLC. Lines 508-9 states that the NLC are in "endocytic vesicles" but no counterstain (eg lysotracker, LAMP etc) has been performed to confirm this so cannot be stated. The subsequent sentence is also ambiguous in that it seems to state something else - that the NLC are not in vesicles but in the cytoplasm itself (rather than in vesicles which are in the cytoplasm?) It is likely that NLCs localise to phagosomes/phagolysosomes in macrophages, which is where MAP has been argued to reside (ref 43 in manuscript). The paper would have been strengthened by determining the subcellular localisation of the NLCs within cells using confocal microscopy together with markers for early/late phagolysosomes.
- Figure 7 - For colourblind readers (as I am - and around 5-10% of readers are) it would be good to have separated channels displayed - or a non-red/green palette. I was unable to interpret this image.
- Line 532 - an "internalized concentration" of 0.78ul/mL is given, but I was unable to find any information on how this was determined. Was it measured directly (e.g. by HPLC?) or estimated - in which case, what are the assumptions? e.g. cell volume? Is the drug expected to distribute from the phagosomes throughout the entire cell? Is the drug effluxed from the cell? This is an important point as the fundamental argument of the manuscript is that delivered drug will be at a concentration that is higher than the MIC of MAP.
- Notes on language/typographical errors
- line 86 - "stablishing", line 436: "stablish" should be changed to "establishing" and "establish" respectively
- line 448 - "preformed" should be "was performed"
- line 451 , 454, 507 - change "have been " to "were ". e.g. "significant differences _were_ found for cell viabilities..."
- Paragraph starting line 445 is somewhat hard to understand/follow and would ideally be reworded. I had to read this several times before I got any sense of what it was trying to convey.
- Sentence starting line 454 "On the other hand..." says that there were statistical differences between RFB loaded NLC and drug in DMSO. However, these differences are not indicated in Fig 5. Have I interpreted this correctly?
Author Response
We would like to thank the reviewers and editor for the careful reading of the manuscript as well as for the positive feedback given. We have tried to answer each concern (our responses in blue) and we think the manuscript has notably been improved.
Comment 1: Section 2.5.3 Macrophage uptake quantification. How/why does this method quantitatively measure "macrophage uptake"? Specifically, what do the authors mean by "uptake"? In normal usage in the field this would be expected to indicate internalisation, but from from the scant description of the assay in the methods section it seems that "cellular/macrophage association" is more likely to be what is being measured. Note that it was very difficult to understand exactly how the assay was performed based on the description in the manuscript. The authors also reference a previous paper from the group (ref 21) but the description in the referenced paper also lacked clarity/detail (for example - the sentence in ref 21 describing "initial amount/[initial fluorescence]" and an internalisation calculation is in garbled English and uninterpretable). Please include a more detailed description of the method and explain how if measures "uptake" rather than "cellular association".
The macrophage uptake quantification assay included in the manuscript attempts to selectively measure NLC uptake by including glycine in the PBS solution employed to wash the cells after the NLC incubation step. This step quench fluorescent signals associated with non-internalized nanoparticles. Moreover, a more detailed description of this assay was included in the methods section. Please, see text highlighted in yellow (section 2.5.3.).
Comment 2: Section 3.1.5 (2.3.6). Experiments were performed to assess stability in the "intracellular environment" of the macrophage using freeze/thaw lysates of cultured macrophages. A whole cell lysate is not a good analogue for the environment that a nanoparticle taken up by a macrophage will encounter. The NLCs will almost certainly be taken up by phagocytosis and will therefore be located in phagolysosomes where the conditions are extreme compared to the general cytoplasmic environment (e.g. highly concentrated reactive oxygen/nitrogen intermediates, degradative enzymes, low pH). Is there any evidence that the RFB loaded into the NLC can survive these conditions in an active form?
We agree with the reviewer with the possibility that NLCs could finally be located within phagolysosomal compartments. However, when conducting a drug release assay from NLC, the use of more biorelevant media is interesting. In this sense, we consider that a macrophage lysate is not a perfect analogue of the NLC environment, but it may be of greater biorelevance than other media described in the scientific literature.
On the other hand, regarding RFB activity after NLC inclusion in phagolysosomal compartments, we think that it would be interesting consider previous published work by Lemmer et al (Lemmer Y, Kalombo L, Pietersen RD, Jones AT, Semete-Makokotlela B, Van Wyngaardt S, et al. Mycolic acids, a promising mycobacterial ligand for targeting of nanoencapsulated drugs in tuberculosis. J Control Release. 2015;211:94-104), which describes the intracellular location of isoniazid-loaded PLGA nanocarriers after their internalization by mycobacterium-infected macrophages. The results reported show that despite nanoparticle localization within phagolysosomes, colocalization phenomena between drug loaded particles and the infecting mycobacteria were reported to occur, suggesting that a therapeutic effect is likely to take place even in the case of nanoparticle inclusion in phagolysosomal compartments.
Comment 3: Section 3.2.2. Confocal microscopy was performed to examine the subcellular localisation of the RFB-loaded NLC. Lines 508-9 states that the NLC are in "endocytic vesicles" but no counterstain (eg lysotracker, LAMP etc) has been performed to confirm this so cannot be stated. The subsequent sentence is also ambiguous in that it seems to state something else - that the NLC are not in vesicles but in the cytoplasm itself (rather than in vesicles which are in the cytoplasm?) It is likely that NLCs localise to phagosomes/phagolysosomes in macrophages, which is where MAP has been argued to reside (ref 43 in manuscript). The paper would have been strengthened by determining the subcellular localisation of the NLCs within cells using confocal microscopy together with markers for early/late phagolysosomes.
We agree with the reviewer that the determination of the sub-cellular location of NLC in macrophages would have enriched the work. Our claim that the nanoparticles were in endocytic vesicles was based on the fact that, as can be seen in confocal images, green fluorescence associated with NLC can be found forming rounded aggregates, that resemble endocytic vesicles. However, it is also true that, as sub-cellular location has not been specifically determined, the exact location of NLC cannot be ensured. Therefore, the paragraph of section 3.2.2. has been modified as follows:
“As observed in Figure 7, both blank (Figure 7.A) and RFB-loaded NLC (Figure 7.B) have been efficiently taken up by macrophages, which constitutes a promising start point for the treatment of infections produced by intracellular pathogens such as MAP [46].”
Comment 4: Figure 7 - For colourblind readers (as I am - and around 5-10% of readers are) it would be good to have separated channels displayed - or a non-red/green palette. I was unable to interpret this image.
Our sincere apologies for not including individual channels in the original manuscript. Separate channels of confocal images were included in the supplementary materials section.
Comment 5: Line 532 - an "internalized concentration" of 0.78ul/mL is given, but I was unable to find any information on how this was determined. Was it measured directly (e.g. by HPLC?) or estimated - in which case, what are the assumptions? e.g. cell volume? Is the drug expected to distribute from the phagosomes throughout the entire cell? Is the drug effluxed from the cell? This is an important point as the fundamental argument of the manuscript is that delivered drug will be at a concentration that is higher than the MIC of MAP.
The internalized concentration was estimated. Following the reviewer's comment, we have realized that we have not consider cell volume in our approach and we have corrected our calculations. In this case, assumptions made were the following: Each well contains 25.000 cells and human macrophages have a cell volume of approx. 4990 µm3, which results in a total cell volume of 1.2475×10-4 mL. On the other hand, a nanoparticle dose of 0.12 mg/mL was administered to each well of THP-1 derived macrophages. Considering that NLC exhibited a drug payload of almost 5%, it could be stated that a total RFB dose of approximately 6 µg/mL was administered to these cells. Therefore, if each well contains 100 µL, then a total RFB amount of 0.6 µg has been administered to cells. Finally, the 13% of internalization reported for macrophages, leads to a total internalized amount of 0.078 µg. Considering this, successfully internalized RFB amount and the above-mentioned total macrophage volume, it is reasonable to think that a intracellular concentration of 625 µg/mL of RFB could be achieved.
We agree with the reviewer that expulsion of the drug from the cell could be a major concern as it could reduce the effective intracellular dose. However, we consider that this would not be the most likely scenario, based on the results of the work by Lemmer et al, 2015 mentioned above. In this work, it has been suggested that the phenomena of colocalization between nanoparticles and mycobacteria could occur through fusion of phagolysosomes. In this way, the RFB would not be free in the cytoplasm and, therefore, it would probably not be the substrate for the efflux pumps that trigger antibiotic resistance.
According to the reviewer suggestions, macrophage uptake assay at the results and discussion section (section 3.2.3.) was modified. Please, see the highlighted text.
Comment 6: Notes on language/typographical errors
line 86 - "stablishing", line 436: "stablish" should be changed to "establishing" and "establish" respectively
line 448 - "preformed" should be "was performed"
line 451, 454, 507 - change "have been " to "were ". e.g. "significant differences _were_ found for cell viabilities..."
Paragraph starting line 445 is somewhat hard to understand/follow and would ideally be reworded. I had to read this several times before I got any sense of what it was trying to convey.
Typographical errors have been revised as requested.
Sorry about this! The paragraph was really confusing. Paragraph starting in line 455 (section 3.2.1) has been reworded. Please, see highlighted text.
Comment 7: Sentence starting line 454 "On the other hand..." says that there were statistical differences between RFB loaded NLC and drug in DMSO. However, these differences are not indicated in Fig 5. Have I interpreted this correctly?
The reviewer has interpreted the text correctly. Statistically significant differences between RFB-loaded NLC and the drug solution have not been correctly indicated in the figure. As a result of reviewer 4, statistics have been changed and Figure 5 has been modified in the manuscript in order to reflect the missing information. Please, see new figure 5 and highlighted sentences related in section 3.2.1.

Reviewer 4 Report
This manuscript needs extensive amendments:
(1) For the speed of centrifugation, g but not RPM should be used.
(2) One Way ANOVA is invalid.
(3) The nano formulation appears to be toxic in CaCo-2 cells (Fig 6). Any clarification and how to avoid such problem in clinical application.
(4) The in vivo data is completely missing.
(5) How is the release in gastric fluid?
(6) How good is the long-term physical and chemical stability?
(7) How good is the safety of the excipients?
Author Response
We would like to thank the reviewers and editor for the careful reading of the manuscript as well as for the positive feedback given. We have tried to answer each concern (our responses in blue) and we think the manuscript has notably been improved.
Reviewer #4:
Comment 1: For the speed of centrifugation, g but not RPM should be used.
As suggested by the reviewer, rpm was replaced by g in the manuscript to indicate the centrifugation speed employed. Please check the modified text in yellow in section 2.3.4.
Comment 2: One Way ANOVA is invalid.
Thanks for your appreciation. We have reflected and realized that a one-way ANOVA is not suitable for the analysis of our results as we have 2 treatments with NLC and 4 concentrations. We have also the experiment carried out with RFB-solutions at equivalent concentrations considering the NLC loading. So, single Figure 5 has been substituted by a new figure with two plots.
We have analyzed the effect of the treatment (Blank or RFB-loaded NLC) and nanoparticle concentration on cell viability through a two-way ANOVA. The two-way ANOVA revealed the existence of statistically significant effects of the treatment, the concentration, and their interaction on cell viability.
Homogeneous subsets pointed out by the post hoc Tukey’s Multiple Comparison Test (p ˂ 0.05) are shown in Figure 5a. Please, see highlighted text and figure 5 in section 3.2.1.
.
Comment 3: The nano formulation appears to be toxic in CaCo-2 cells (Fig 6). Any clarification and how to avoid such problem in clinical application.
Figure 6 shows the cytotoxicity of the developed formulations in THP-1-derived macrophages. Despite the toxicity exhibited by the NLC at high concentrations (e.g. 0.3 mg/mL), the administration of a nanoparticle dose of 0.12 mg/mL led to a cell viability ≥ 70%, which has been reported to be the threshold, according to ISO 10993-5 where a cytotoxic effect is believed to occur. Therefore, we have selected this concentration to proceed with further assays.
Comment 4: The in vivo data is completely missing.
This work is aimed to evaluate the potential of RFB-loaded NLC. Because of that, and considering the principle of the 3Rs (replacement, reduction, and refinement) that should guide animal experimentation, in this initial stage of formulation development we have decided using in vitro evaluation of the nanosystems in cell lines, such as THP-1 derived macrophages and Caco-2 cells instead of in vivo experimentation. Nonetheless, due to the promising results obtained, and keeping in mind that in vivo studies are undoubtedly essential in drug delivery systems testing, future steps of the work could require studies with infected macrophages followed by in vivo studies.
Comment 5: How is the release in gastric fluid?
Other researcher’s results show that NLCs are unstable in acidic conditions, such as gastric fluid. (Ana R, Mendes M, Sousa J, Pais A, Falcao A, Fortuna A, et al. Rethinking carbamazepine oral delivery using polymer-lipid hybrid nanoparticles. Int J Pharm. 2019;554:352-65). Oral administration of these NLCs would require their inclusion in gastro-resistant pharmaceutical forms, either in capsules or included in enteric-coated multiparticulate systems. We have therefore ruled out performing a release assay in gastric fluid.
Comment 6: How good is the long-term physical and chemical stability?
The stability of the developed NLC after 1 month of storage at 5±1° , was addressed in our previous paper (Rouco H, Diaz-Rodriguez P, Rama-Molinos S, Remuñán-Lopez C, Landin M. Delimiting the knowledge space and the design space of nanostructured lipid carriers through Artificial Intelligence tools. Int J Pharm. 2018;553 (1-2):522-30). The stability has proven to be adequate in terms of particle size, polydispersity index and drug payload. Minor changes were found for zeta potential during the storage period, which are not likely to affect colloidal stability of NLC. This information relative to NLC stability has been added in the section 3.1 (Please, see highlighted text).
To avoid stability problems over longer periods of time, conversion of the NLC dispersions into solid dosage forms would be necessary. Anticipating these issues, this work includes the thermal evaluation of NLCs, in order to establish their ability to withstand, without altering, techniques such as spray drying, which is useful to obtain solid powders from these NLC dispersions.
Comment 7: How good is the safety of the excipients?
Main components of NLCs are Precirol ATO 5 (GRAS, generally regarded as safe) and oleic acid. Despite some authors have reported certain cytotoxic effects with the use of oleic acid (Yin, H.; Too, H.P.; Chow, G.M. The effects of particle size and surface coating on the cytotoxicity of nickel ferrite. Biomaterials 2005, 26, 5818-5826) or the emulsifiers that we have been used in this work, the cytotoxicity of the bulk materials has not been evaluated as biocompatibility of the nanoparticles was experimentally determined and has proven to be adequate

Round 2
Reviewer 2 Report
I think the current revised version seems OK for me
Reviewer 4 Report
The reviewer has no objection to accept this manuscript.